# Dynamics of pH at Santa Catalina Island

**Craig G. Gelpi**⦾*

Catalina Marine Society, Lomita, California, United States of America

* craig@catalinamarinesociety.org

## Abstract

The local expression of ocean acidification may depend on local oceanographic features in addition to global forcings. Our objective is to provide a baseline of pH behavior at Santa Catalina Island, situated within the unique oceanographic characteristics of the Southern California Bight, and to gain insight into ocean acidification at the island. Measurements of the upper water column (to 30-m depth) of pH, temperature, conductivity, chlorophyll and dissolved oxygen at Santa Catalina were made from a fixed mooring and by profiling the water column from a boat and on Self-Contained Underwater Breathing Apparatus (SCUBA). The average pH (8.095 at 18-m depth) was found to be higher than that reported off the nearby mainland and the Northern Channel Islands. The higher value is thought to result from both downwelling produced by internal waves as well as less upwelling at the island compared to other locations. Large modulations in pH at depth corresponded to advection of gradients by internal waves. Within the accuracy of the sensors there was no seasonal dependence detected at near-surface, nor a pH signal associated with the sub-surface chlorophyll and oxygen maxima. We conclude that marine life living at depths affected by internal waves experience significant variation in pH.

## Introduction

Global ocean acidification is recognized as a stressor to marine ecological systems and is increasingly being studied and quantified. How ocean acidification is expressed locally will inform local responses to protect, accommodate or remediate oceanic waters [1]. Because ocean conditions differ substantially between the Southern California Bight (SCB) and the Central and Northern California coasts [2], we expect that ocean acidification will be manifested differently in the bight as compared to other regions in the California Current [3]. The Bight itself is internally complex, with eight islands, a recirculation arm of the California Current, anthropogenic sources of nutrients, upwelling patterns and wind conditions that differ between the Northern and Southern Channel Islands. Hence, we expect the ocean acidification expression also to differ within the bight.

The Catalina Dynamic Ocean Chemistry (CDOC) project uses opportunistic measurements of chemical quantities off the lee side of Santa Catalina Island, California, to understand similarities and differences among the Channel Islands within the Southern California Bight and nearby mainland coast including the Los Angeles metropolitan area. In addition to pH, the measurements usually include temperature, conductivity, dissolved oxygen and

**Data Availability Statement:** All relevant data are within the paper and its Supporting information files.

**Funding:** The author received no specific funding for this work.

**Competing interests:** The author has declared that no competing interests exist.

chlorophyll. Data were collected from the upper water column (to 30-m depth) with the objectives to determine mean levels and variability of pH at Santa Catalina. The results will be useful for understanding how ocean acidification may affect the island.

Ocean acidity at specific locations can vary for many reasons, depending on sources and sinks as well as advection of waters with different pH levels. Hofmann, *et al.* [4] reviewed pH measurements made in several different environments over the globe to conclude that pH can vary significantly depending on environment. In addition to the global source of increasing atmospheric $CO_2$ levels, local changes in $CO_2$ via respiration by marine fauna and uptake by phytoplankton and macroflora result in changing pH. Advection of pH gradients via wind-induced upwelling, bringing lower pH waters to the surface, internal-wave induced upwelling or downwelling advecting gradients at depth, or changes in the percentages of source water (at Santa Catalina, source waters are Alaskan or Equatorial waters) can also modulate pH values at specific California locations. We expect phytoplankton growth to increase pH as it is a sink for carbon dioxide, upwellings should decrease pH and lateral advection of Pacific Equatorial waters into the bight surrounding Santa Catalina Island should decrease it as well [5, 6]. Hofmann, *et al.* [4] also discuss the need for pH data specific to a given species' natural habitat to enable understanding of ocean acidification on it. Hence, our studies may be applied to Santa Catalina's unique assemblage of marine life [7].

There have been several published studies of pH measured either along the Northern Channel Islands or adjacent to the California mainland in the bight to which we can compare our results. These include: Kapsenberg and Hofmann [8] who examined time series of pH around the Northern Channel Islands and Friedler, *et al.* [9] who measured both time series and spatial variability within a kelp forest off San Diego, California.

## Materials and methods

Typically, pH, temperature, and conductivity data were collected on deployments and often chlorophyll and dissolved oxygen were collected, too. The data were measured with sensors produced by YSI and attached to a YSI EXO sonde. The sensors, their accuracies and resolutions are listed in Table 1. The sensors were calibrated the week before deployment and often within a week after retrieval. Calibration and maintenance were performed according to the manufacturer's guidance and for pH involved using standard buffers of 7 and 10. The pH sensor is of the ion specific electrode (ISE) type, responsive to $H^+$. An analysis of post calibration results indicates a bias toward higher pH and is described in S1 File. The chlorophyll sensor was not calibrated so only relative changes during a deployment are considered.

There was no attempt to equilibrate the pH measurements for temperature. All other factors being equal, proton activity will increase with increasing temperature. Both extrapolating the temperature dependences of the calibration buffers to the pH of sea water and measuring pH on the same sample of sea water while periodically cycling its temperature indicated a temperature dependence of -0.007 pH unit/˚C.

**Table 1. Measured parameters and accuracies.**

| Sensor | Accuracy |
|---|---|
| Temperature | ± 0.2˚C |
| Conductivity | ±1.0% |
| pH | ±0.1 pH unit |
| Dissolved Oxygen | ±0.1 mg/l or 1% reading |
| Chlorophyll | Not calibrated |

The optical dissolved-oxygen sensor was calibrated using an air-saturation method, while the conductivity was calibrated using a conductivity standard. The conductivity accuracy was too coarse for absolute measurements to be useful, but relative changes in conductivity and derived salinity are useful for correlating with other parameters. When displayed, we scaled measured salinity to agree in the mean with that of the nearest California Cooperative Oceanic and Fisheries Investigation (CalCOFI, www.CalCOFI.org) measurements at the same depth, made twice a year to the east of the island.

A site map, shown as the satellite-measured sea-surface temperature (SST) averaged over the year 2019 [10] is shown in Fig 1. SST is chosen to facilitate a later discussion. All CDOC sonde deployments were on the submarine slopes of Santa Catalina Island, near Avalon or Two Harbors, marked A and T on Fig 1, respectively. The sondes were deployed by volunteer boaters and divers and deployment dates were determined by volunteer availability. Deployments were of two types: depth fixed, where the sonde is attached to the Dirk Burcham Memorial Scientific Mooring chain at a fixed depth; and, depth profiling, where the sonde sampled the water column as it was lowered from a boat or carried by a SCUBA diver. All deployments were within a few hundred meters of a *Macrocystis pyrifera* kelp forest, and SCUBA deployments traversed a kelp forest while collecting data.

The mooring was located in the Wrigley Marine Reserve (aka, WIES) outside of Two Harbors (33.4469 ˚N, 118.4888 ˚W, marked as "T" in Fig 1), roughly 30 km northwest of Avalon. SCUBA divers fixed the sonde at 18.3-m (60 ft) depth to a housing attached to a mooring chain where the seabed is approximately 30-m deep. In addition, temperature recorders were attached to the chain at 6.1-m (20 ft), 12.2-m (40 ft) and 24.4-m (80 ft) depths, straddling the sonde depth. All fixed-depth sensors sampled at 30-min intervals. The seven fixed-depth deployments are listed in Table 2 and typically had a duration of 2 to 3 weeks. The sonde on the mooring was equipped with a brush that wipes the sensor heads every 30 minutes to mitigate biofouling. Upon retrieval, the sensors were inspected for biofouling but nothing unusual was noted on the sensor heads, though growth often was seen on the housing.

The depth-profiling procedure has the sonde sampling every 20 seconds while it is lowered to 30 meters, stopping for 5 minutes at 6.1, 12.2, 18.3, 24.4 and 30.5 (100 ft) meters, and reversing the procedure upon bringing the sonde to the surface. For this study, we only used the ascending data to ensure the sensors equilibrated with ocean temperatures. When deployed by SCUBA, the sonde was transported along the seabed gradient and held for 5 minutes at the

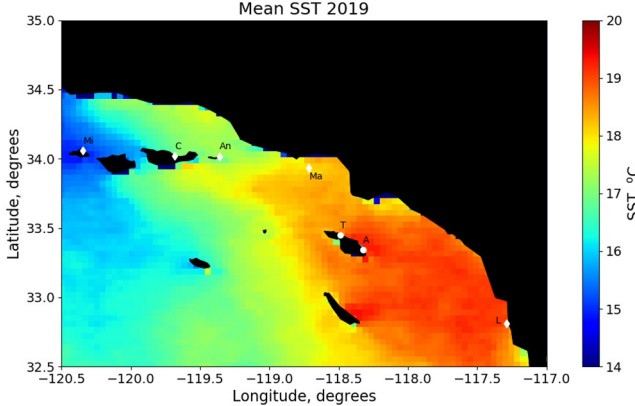

**Fig 1. Site map.** Mean 2019 SST map of SCB. Marked locations correspond to key in Table 6: white dots are CDOC and diamonds are sites referenced.

**Table 2. Mooring deployments.**

| Date (Day of Year) | Thermograph depths, m |
|---|---|
| 07/14/2018-08/02/2018 (204) | 6.1, 12.2, — |
| 09/20/2018-10/12/2018 (274) | 6.1, 12.2, 24.4 |
| 12/22/2018-01/05/2019 (363) | 6.1, 12.2, 24.4 |
| 03/16/2019-03/30/2019 (82) | 6.1, 12.2, 24.4 |
| 06/01/2019-06/15/2019 (159) | 6.1, 12.2, 24.4 |
| 12/28/2019-01/12/2020 (5) | 6.1, —, 24.4 |
| 02/08/2020-03/08/2020 (53) | 6.1, 12.2, 24.4 |

seabed at the above depths, with the exception of the 30.5-m station, which was not observed due to safety considerations. The list of 29 depth-profiling deployments (27 collecting pH data) is shown in Table 3. Most of the depth profiles (and all the SCUBA deployments) were made outside Avalon, near the coast guard buoy (CGB in Table 3, at 33.343°N, 118.3251°W, marked as "A" in Fig 1) where the seabed depth is 45 m. Two depth-profiling deployments, at

**Table 3. Depth-profiling deployments.**

| Date (DOY) | Location | Comments |
|---|---|---|
| 6/17/2017 (168) | CGB | Coast Guard Buoy |
| 8/20/2017 (232) | CGB | |
| 10/4/2017 (277) | CGB | |
| 12/29/2017 (363) | CGB | |
| 1/21/2018 (021) | CGB | no pH |
| 2/25/2018 (056) | CGB | no pH |
| 3/18/2018 (77) | CGB | |
| 4/29/2018 (119) | CGB | |
| 5/20/2018 (140) | Two Harbors | |
| 6/16/2018 (167) | CGB | |
| 9/9/2018 (252) | Emerald Bay | To 24.4 m |
| 11/17/2018 (321) | CGB | |
| 3/24/2019 (083) | CGB | |
| 5/26/2019 (146) | CGB | bad conductivity |
| 9/01/2019 (244) | CGB | |
| 1/01/2020 (001) | CGB | Temperature and pH only |
| 1/11/2020 (011) | CGB | SCUBA to 24.4 m |
| 2/16/2020 (047) | CGB | |
| 3/07/2020 (067) | CGB | SCUBA to 24.4 m |
| 6/22/2020 (174) | CGB | pH, no temperature |
| 7/24/2020 (205) | CGB | |
| 8/20/2020 (233) | CGB | |
| 9/25/2020 (268) | CGB | |
| 3/14/2021 (73) | CGB | SCUBA to 24.4 m |
| 6/27/2021 (178) | CGB | |
| 8/22/2021 (234) | CGB | |
| 12/12/2021 (346) | CGB | |
| 1/08/2022 (8) | CGB | |
| 1/23/2022 (23) | CGB | SCUBA to 24.4 m |

Emerald Bay and Two Harbors, were approximately a kilometer from the fixed-depth mooring deployments ("T" in Fig 1). No permits were required as the measurements were made at public access locations with transitory, nonattached, sensors.

The two sampling protocols have relative advantages and disadvantages. The advantage of fixed-depth measurements is that the instrument can be unattended for long periods of time, collecting much data and thereby enabling better analyses with respect to statistics and dynamic behavior. Depth dependences of parameter values obtained from the moored sonde are inferred from the correlation of the parameters with temperature as described in the analysis section. The advantages of depth-profiling include: direct sampling of the water column and more deployments are possible because less-specialized personnel are required to deploy the sonde from a boat. Because measurements from each deployment are expected to have independent bias uncertainties remaining after calibration, more deployments mitigate bias effects by averaging.

Analysis consisted of inspection of times series to determine mean and extreme values, calculating correlations amongst measured parameters, determining depth gradients of the measured parameters, inspecting for seasonal behavior, and calculating power spectral densities and spectral coherences via Fourier transform for frequency analysis.

Analysis of calibration results, described in the S1 File, indicates a positive bias in the pH measurements. Accordingly, we subtract 0.04 pH units from statistical means of pH, and note when the bias correction is applied.

## Results

### Fixed-depth deployments

Analysis and results are presented separately for the two deployment protocols, as the analysis procedures differed between the two. First, we describe the fixed-depth deployment data analysis. Example time series for the temperature, pH and salinity data are shown in Fig 2 when the first measurements (July 2018) were made during strong summer stratification conditions with the mixed-layer depth on the order of 5 meters. Time is referenced from the beginning of the deployment. Temperature was measured at 6.1 m and 12.2 m with temperature loggers and at 18.3 m on the sonde for the summer deployment. For later deployments, an additional thermograph was placed at 24.4 m. In Fig 2, the most dominant features are upwelling and downwelling produced by apparent internal waves, which have been well studied in this region [11, 12]. The largest modulations are at depth, where the temperature varies from a low ~13 °C

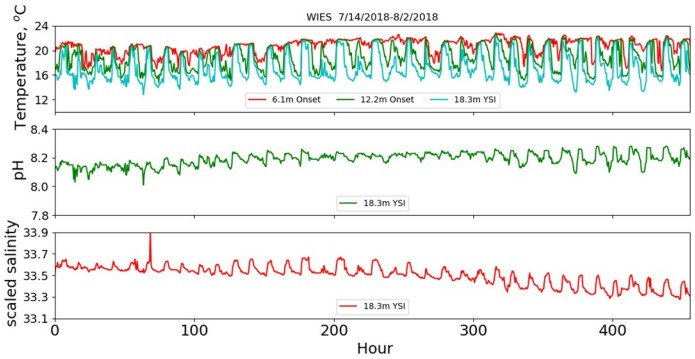

**Fig 2. July 2018 mooring deployment.** Time series of temperature, pH and scaled salinity for the 7/14/2018 mooring deployment.

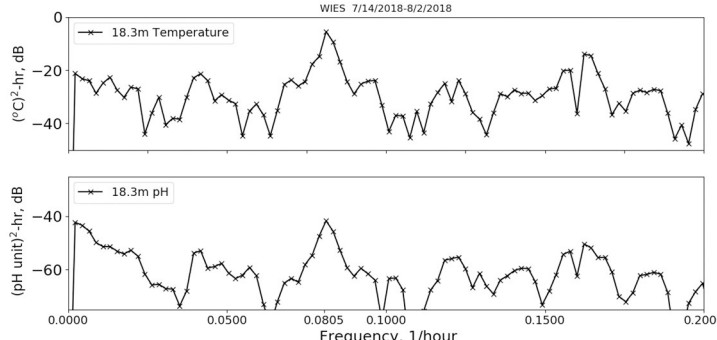

**Fig 3. Power spectral densities.** Power spectral density for temperature (top) and pH (bottom) for the 7/14/2018 deployment.

up to the near-surface value of ~20 °C. Fast Fourier-transform power spectral density calculations on the 460-hour summer time series, shown in Fig 3, indicate a peak in power at 0.0811 hr$^{-1}$, which is within a Fourier transform frequency bin (0.002 hr$^{-1}$) of the $M_2$ lunar tidal cycle (0.0805 hr$^{-1}$), consistent with internal wave kinematics as studied by Gelpi [12]. From Fig 2, we find that the temperature, pH and salinity variations are well correlated at depth at semi-diurnal frequencies. This is shown also in Fig 4, where the coherences between temperature and: pH, salinity, dissolved oxygen (DO) and chlorophyll (Chl) as a function of frequency are plotted. The coherence was computed with a 5-frequency-bin smoothing, yielding an expected coherence value of 0.2 for random noise. The pH-temperature coherence is 0.99 at 0.0811 hr$^{-1}$, with a phase (not shown) of 9°, i.e., in phase, showing that the pH decreased with decreasing temperature, opposite the trend produced by changing ionic activity. The correlation of salinity with temperature is also near 1 at semi-diurnal frequencies and implies that the salinity gradient was directed toward the surface. This is consistent with glider results taken near the island [13], which report a subsurface salinity minimum at similar depths. Also, an examination of CalCOFI data (shown in the supporting material) taken near this time and location, also shows an upper-water-column gradient produced by higher salinity values near the surface. The coherences between temperature and dissolved oxygen and Chl are less, about 0.8 at

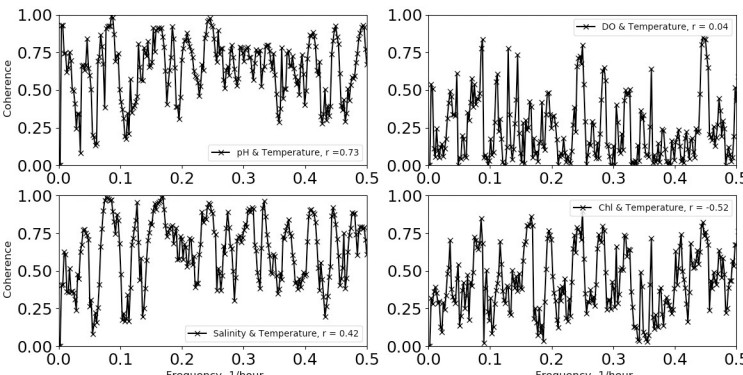

**Fig 4. Temperature coherences and correlations.** Coherence between pH and temperature (top left), salinity and temperature (bottom left), DO and temperature (top right) and Chl and temperature (bottom right) for the 7/14/2018 mooring deployment. Applicable correlation coefficients are shown in the legends.

0.0811 hr$^{-1}$. In addition to the coherences, the correlation coefficient, r, between the various time series are given in the legend for each plot. The correlation between pH and temperature is 0.73 and is much higher than the correlations between temperature and DO (r = 0.04) and Chl (r = -0.52). For completeness, the coherences and correlations between pH and Chl, salinity and DO are shown in Fig 5, as well as DO and Chl. We note that DO and pH have a higher coherence (top right panel of Fig 5) at the sum of the diurnal and semidiurnal frequencies, namely, 0.125 hr$^{-1}$, but referring to Fig 3, there is no significant power in pH at this frequency.

In addition to the internal-wave modulations, there are broader trends in pH and salinity that exhibit an anti-correlation relationship, with salinity decreasing by 0.2 psu as pH increased by 0.1 over the nearly three weeks of the deployment (Fig 2). The correlation at semi-diurnal frequencies and the anti-correlation at very low frequencies explains the low correlation coefficient between pH and salinity (bottom left panel of Fig 5). During this time, near-surface temperature trended upward as expected because the seasonal surface temperature maximizes in August [11]. This was the only mooring dataset that exhibited a broad trend in salinity and pH over the deployment duration. We note here that the trend is consistent with high pH, low-salinity California Current waters replacing lower pH, warmer Pacific Equatorial waters as described by Meinvielle and Johnson [5] and Nam, *et al.* [14] superposed on seasonal warming.

Chlorophyll and dissolved oxygen did not vary simply with temperature. Scatter plots of these quantities versus temperature are shown in Fig 6 for the 07/14/2018 deployment, as well as cubic-polynomial fits as a function of temperature. Both chlorophyll and dissolved oxygen exhibit maximum values at temperatures nominally found at deeper depths. The average temperatures at 6.1 m, 12.2 m and 18.3 m are 20.8˚, 18.8˚ and 17.0˚ C, respectively, while the peak chlorophyll values (actual and from the fit) correspond to 16.0˚ C, which from extrapolation of the temperature values would nominally be at 21-m depth, that is, below the sonde. The dissolved-oxygen values also peak at temperatures that are nominally below the sonde.

Fig 7 is the scatter plot of pH versus temperature superposed on the line representing the quadratic fit. The fit suggests that pH has a greater correlation to temperature at lower temperature values as the correlation decreases at higher temperatures. The overall pH does not exhibit a maximum at intermediate temperatures as the chlorophyll and dissolved oxygen do (Fig 6).

In contrast to the highly-stratified summer water, the well-mixed (the mixed-layer depth was often below 24 m) winter case shown in Fig 8, did not exhibit modulations at typical tide-

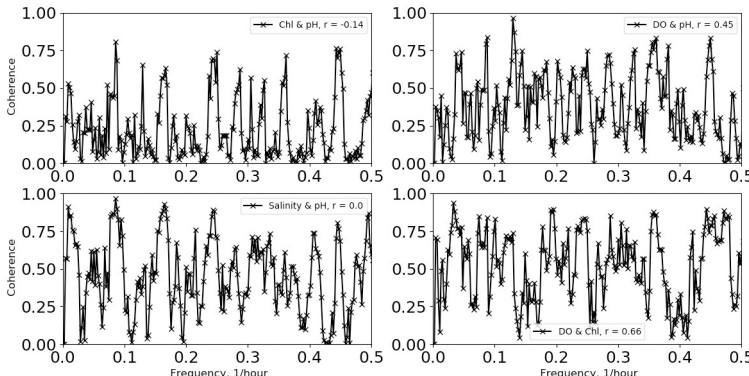

**Fig 5. pH coherences and correlations.** Coherence between Chl and pH (top left), salinity and pH (bottom left), DO and pH (top right), and DO and Chl (bottom right) for the 7/14/2018 mooring deployment. Applicable correlation coefficients are shown in the legends.

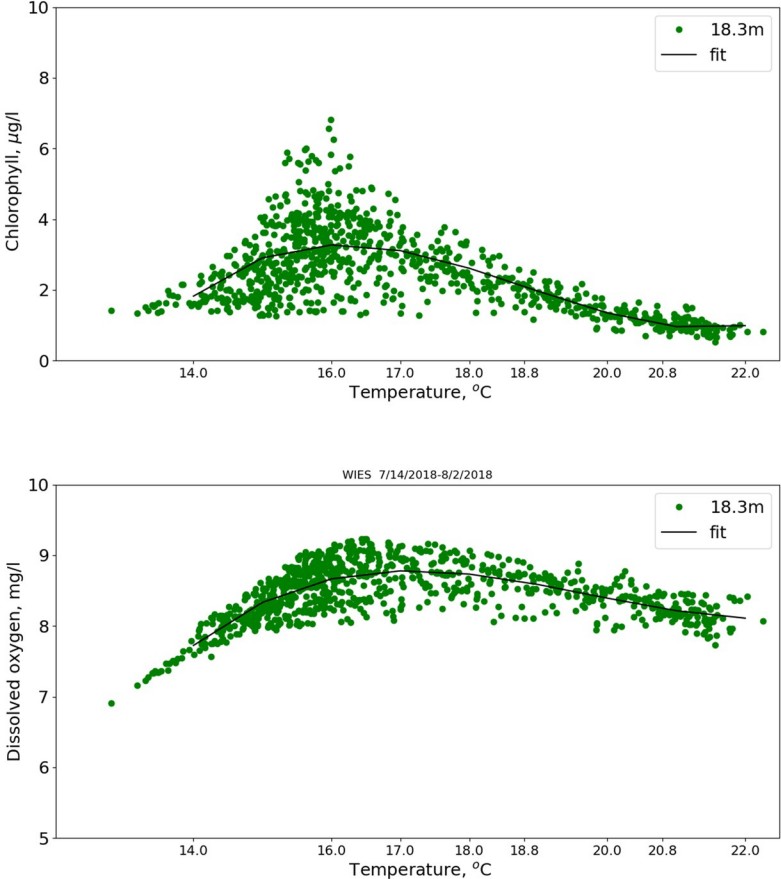

**Fig 6. Chlorophyll and oxygen 7/2018.** Chlorophyll (top panel) and dissolved oxygen (bottom) versus temperature for July, 2018 deployment.

driven internal-wave frequencies. Occasionally, there were cold-water intrusions found at 24.4 m and sometimes shallower. These upwellings were accompanied by pH and salinity modulations again correlated with temperature changes, as well as modulations in chlorophyll and dissolved oxygen (not shown). The upwelled water also exhibited lower salinity, consistent

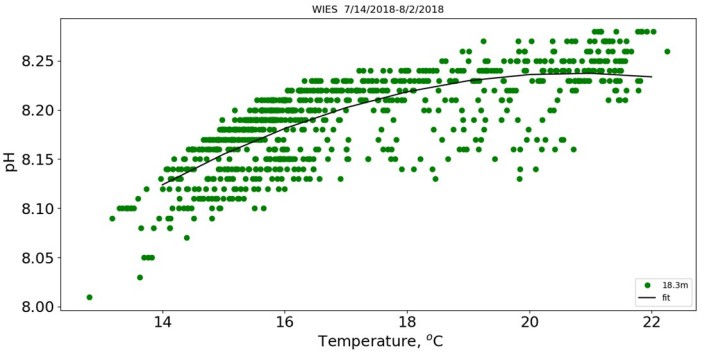

**Fig 7. pH 7/2018.** pH vs temperature for the July 2018 deployment.

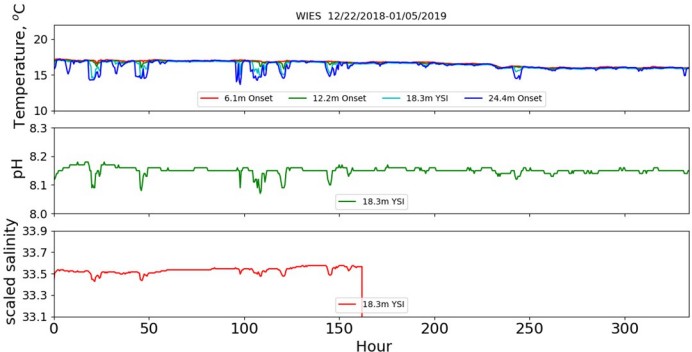

**Fig 8. December 2018 mooring deployment.** Time series of temperature, pH and scaled salinity for the 12/22/2018 mooring deployment.

with the summer results and CalCOFI data. The salinity values for this deployment were judged problematic beyond hour 160 and are not shown past that time. The general qualitative characteristics found in the stratified data also can be found in the well-mixed case, but because there is relatively little temperature modulation and the chlorophyll values are much smaller, the phenomena are not pronounced. The broad trend to lower temperature is expected as near-surface temperature approached the seasonal minimum in March [11].

These two cases represent the extremes in dynamical behavior found in the seven fixed-depth deployments, with the mentioned broad trends in pH and salinity found in the July 2018 data not being found in the other datasets. The mean values of temperature, dissolved oxygen and pH were computed and are shown in Table 4 for all fixed-depth deployments. Also given is an estimate of pH, $pH_{est,}$ computed from proxies as described in a later section.

Although the sonde was at a fixed depth, we were able to measure parameters as a function of nominal depth using the phenomenon of internal waves. As the internal-wave height changed at the mooring, water originating from deeper or shallower depths as shown by the temperature changes was advected passed the sonde. To compute the depth dependence a second-degree polynomial was fit to the data for each deployment, yielding pH as a function of temperature. The average temperature for each thermograph depth was determined for each deployment, and that temperature was used to retrieve the nominal pH, $\widehat{pH}$, from the fit. Or,

**Table 4. Average values at 18.3 m from fixed-depth deployments.**

| Date | Temperature ˚C | O$_2$ mg/l | pH, measured | pH$_{est}$ |
|---|---|---|---|---|
| 07/14/2018-08/02/2018 | 16.9 | 8.46 | 8.19 | 8.08 |
| 09/20/2018-10/12/2018 | 18.3 | 7.78 | 8.21 | 8.07 |
| 12/22/2018-01/05/2019 | 15.6 | 7.80 | 8.14 | 8.02 |
| 03/16/2019-03/30/2019 | 14.3 | 8.11 | 8.22 | 8.01 |
| 06/01/2019-06/15/2019 | 15.2 | 7.93 | 8.12 | 8.02 |
| 12/28/2019-01/12/2020 | 15.5 | 8.02 | 8.17 | 8.03 |
| 02/08/2020-03/08/2020 | 15.6 | 8.67 | 8.11 | 8.07 |
| Average | | | 8.17 | 8.04 |
| [Bias corrected] | | | [8.13] | |

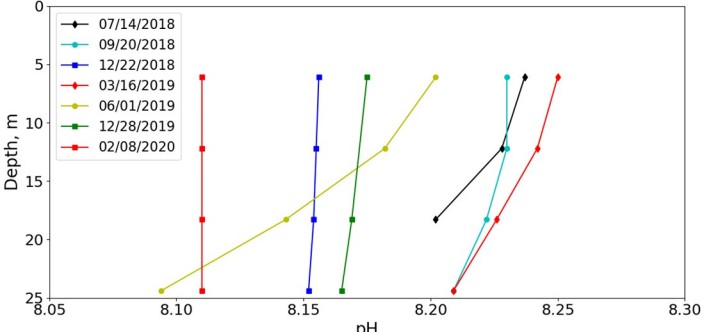

**Fig 9. Inferred depth dependence.** Inferred depth dependence of pH for each deployment.

in equation form:

$$\widehat{pH}(z) = pH(\langle T(z) \rangle)$$

Here, $T(z)$ is the temperature measured at $z$ (i.e., measured by the instrument at $z$), the ensemble bracket denotes the time-averaged temperature at $z$, and $pH(T)$ is the fit of pH to temperature.

The nominal or inferred pH-depth dependences for all fixed-depth deployments are shown in Fig 9. The inferred pH is seen to decrease with depth for all deployments. However, under well-mixed conditions, such as occurred during the December 2018 deployment (Fig 8) and other winter deployments, the pH gradient was relatively small.

## Depth profiling

Depth profiling probes the water column directly rather than letting water characteristics of nominal depths sweep pass the sonde. However, internal waves will contribute to the variability of the depth profiles, as logistically each profile was limited to approximately an hour, essentially measuring a single random phase of the dominant semi-diurnal-frequency internal waves. As an example, the temperature, pH, chlorophyll and dissolved oxygen data for one profile (executed 4/29/2018) are shown in Fig 10. The upper 18 m exhibit nearly constant temperature but a discrete thermocline is found at 18 m. Below 18 m, the temperature decreases linearly with depth. At 18 m, dissolved oxygen also exhibits a discrete change increasing from

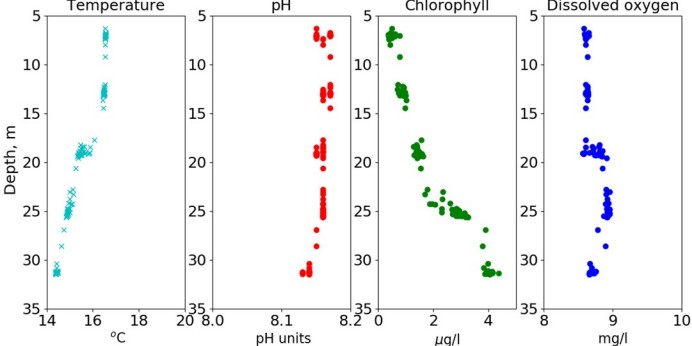

**Fig 10. April 2018 depth profile.** Example of depth-profiling data, executed 4/29/2018.

8.5 mg/l to nearly 9 mg/l below the thermocline. The chlorophyll maximum is at or below the deepest depth sampled, while the dissolved oxygen maximum appears to be above the chlorophyll maximum, similarly to what is implied for the fixed-depth deployment shown in Fig 6. The pH values are relatively constant for depths between 6 and 25 m, toggling by the sensor quantization, and decreasing below 25-m depth.

The data for each deployment within a 2-m range of a deployment depth station were averaged to form a nominal depth profile. The temperature and pH profiles for every depth-profiling deployment are shown in Fig 11, with the color and symbol coding reflecting the date of data collection. Profiles corresponding to apparent complete mixing (i.e., no temperature

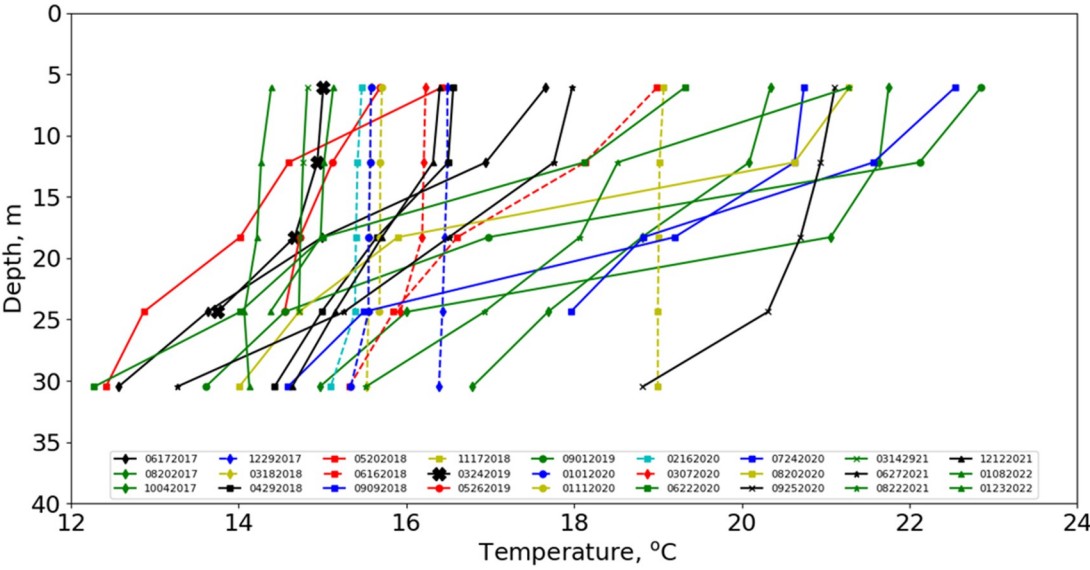

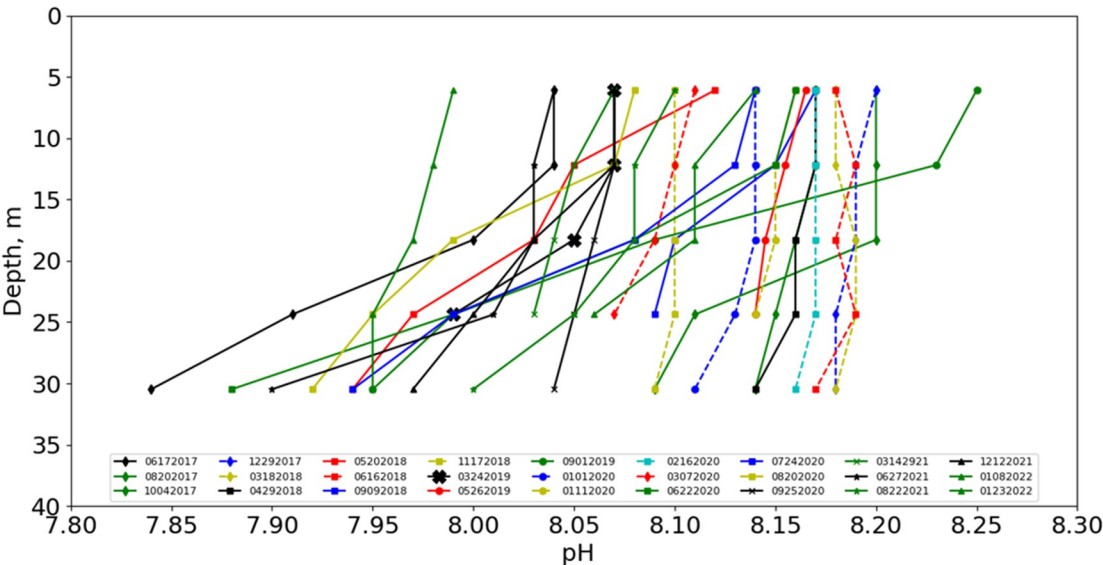

**Fig 11. Depth profiles.** Temperature (top) and pH (bottom) profiles from the depth-profiling protocol.

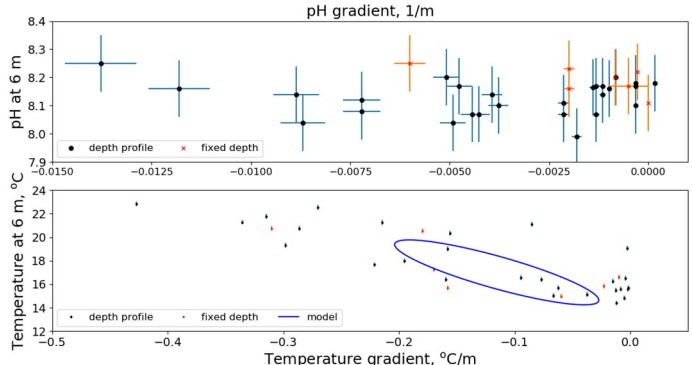

**Fig 12. Surface values vs gradients.** Surface value vs. gradient, pH (top) and temperature (bottom).

stratification) are coded with dashed lines, the others with solid lines. Note that when there was little stratification, the temperature tended to be between extremes values of the stratified cases.

The pH profiles for the non-temperature-stratified cases also show little variation with depth. Note that pH was not measured on every depth profile (see Table 3). In general, larger pH depth gradients correspond with greater temperature stratification. The largest gradients correspond to a change of almost 0.3 pH units from 6.1 m to 30.5 m (0.0122 pH unit/m).

In contrast to the temperature profiles, the pH profiles exhibited no correlation between near-surface values and gradients. We make this explicit by plotting in Fig 12 the values measured at 6-m depth versus the gradient. The gradient was computed using a first-order polynomial fit to the depth-profiling data. Mooring data are also included in Fig 12 using the nominal inferred pH data shown in Fig 9. Also shown in the bottom panel of Fig 12 is the 6-m temperature vs the temperature gradient. In addition to the data, the 6-m temperature versus gradient from our temperature model is shown. The model was developed mainly from temperatures measured at Santa Catalina during the 1990s [15]. The locus of model points (i.e., the ellipse) is produced by the seasonal dependence of the temperature, temperature gradient and their phase difference. Temperature has a well-defined relationship between near-surface values and depth gradients in both the model and in the data. Because the model represents data that were substantially averaged (almost a decade of hourly samples) larger variations as found in the CDOC data are not reproduced in the model. In the pH data no relationship between surface pH and its gradient is found. The average and standard deviation among deployments of pH values for each depth station are listed in Table 5. At 18.3 m the average pH is 8.06. Using values from the table we compute a gradient of 0.004 pH unit/m.

**Table 5. Average pH among depth-profiling deployments.**

| Depth, m | Average pH (bias corrected) |
|---|---|
| 6.1 | 8.088 ± 0.060 |
| 12.2 | 8.078 ± 0.062 |
| 18.3 | 8.057 ± 0.066 |
| 24.4 | 8.025 ± 0.083 |
| 30.5 | 7.989 ± 0.113 |

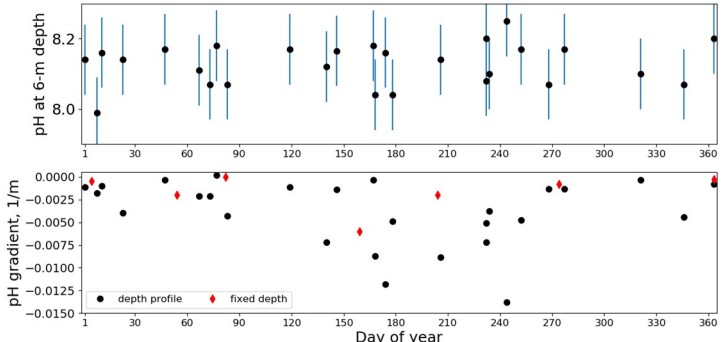

**Fig 13. Day-of-year pH.** Near-surface (top) and gradient values (bottom) of pH organized by day of year.

The temperature, chlorophyll and pH data were also examined with respect to day of year (DOY). The temperature variation was consistent with our models [15]. The chlorophyll variation (not shown) was inconsistent with the average chlorophyll derived from the satellite remote-sensing product from MODIS [16]. The average remotely-sensed chlorophyll was largest in late winter. Instead, large values of chlorophyll were found in the summer and probably related to internal waves advecting high-chlorophyll-content water from depth. Near-surface pH and the pH gradient are plotted versus DOY in Fig 13. As can be inferred from Figs 12 and 13 indicates there is no statistically discernible pattern in the near-surface pH. However, the gradient does exhibit a pattern of small gradients during the late fall and winter, and much more variable gradients during the spring and summer. Of course, this is the time when internal waves are expected to add much scatter to the depth-profiling measurements.

## Comparison with other bight measurements

In this section, we describe published results of pH measured around the Southern California Bight and organize them into a table for easy comparison with CDOC results. Frieder, *et al.* [9] have executed measurements perhaps closest to CDOC in process, near the mainland shore in the La Jolla Kelp Forest (32.81˚N, 117.29˚W, site "L" in Fig 1). Using two sets of instrumentation, they successively measured along-shore, cross-shore and depth gradients in pH and other chemical parameters on deployments of 3-month durations. To determine the depth gradient, they measured pH simultaneously at 7-m and 17-m depths and found mean pH to be 8.07 and 7.87, respectively, i.e., a gradient of 0.02 pH unit/m. These compare to our bias-corrected depth-profile values of 8.09 at 6.1 m and 8.06 at 18.3 m. Hence there is a significant difference of 0.19 between the two experiments at the deeper depth. They also found large depth gradients in pH when the water column was stratified with the near-surface measurements being stable. This is similar to CDOC findings. The average of their along-shore measurements (8.05 in the north and 8.10 in the south, both at 7 m) is 8.075. This is a smaller value, though within the uncertainty, for the similar depths shown in Table 5.

Kapsenberg and Hofmann [8] report pH measurements made off the Northern Channel Islands, specifically, Anacapa ("An", 34.0164˚ N, 119.362˚ W, 3–4 m depth), Santa Cruz ("C", 34.0204˚ N, 119.6843˚ W, 3–4 m depth, also) and San Miguel ("M", 34.057˚ N, 120.3455˚ W, 6-m depth). The Anacapa and Santa Cruz instruments were located in a kelp forest and eel-grass bed, respectively, providing biological activity expected to influence pH through photosynthesis and respiration. Averages for the 3 years of data are: Anacapa, 8.01±0.04; Santa Cruz,

**Table 6. pH comparisons.**

| Reference, location, map key and depth | pH, (extrapolated to 18 m) |
|---|---|
| Frieder, et al. [9]: La Jolla KF (L), 17 m | 7.87 (7.86) |
| Kapsenberg and Hofmann [8]: San Miguel (Mi), 3–4 m | 8.05 (7.88) |
| Kapsenberg and Hofmann [8]: Santa Cruz (C), 3–4 m | 8.00 (7.83) |
| Kapsenberg and Hofmann [8]: Anacapa (An), 6 m | 8.01 (7.87) |
| Leinweber and Gruber [17]: SMBO (Ma), upper 20 m | 8.08 |
| Alin, *et al.* [18]: SoCal Bight, 20 m | 8.03 (8.05) |
| Alin, *et al.* [18]: applied to CDOC Temp and $O_2$ at 18 m | 8.04 |
| CDOC fixed-depth (T), bias corrected, at 18 m | 8.13 ± 0.04 |
| CDOC depth-profiling (A), bias corrected, at 18 m | 8.06 ± 0.07 |

8.00±0.06; and San Miguel, 8.05±0.05. These values are significantly smaller than those for the closest depth in Table 5.

Leinweber and Gruber [17] calculated pH from dissolved inorganic carbon (DIC), alkalinity, temperature and salinity measured at the Santa Monica Bay Observatory mooring located at 33.9317˚N, 118.715˚W (marked as "Ma" in Fig 1), which is 6 km from the mainland and 31 km from our Two Harbor mooring. They determined the median pH for the upper 20 m to be 8.08. This value compares well with the average pH among the 3 depths above 20 m in Table 5, i.e., 8.07.

Alin, *et al.*, [18] developed empirical relationships between pH, temperature and dissolved oxygen using data from a calibration cruise along the west coast, between Monterey Bay and Punta Eugenia, Mexico, that is, including a line of stations within the Southern California Bight. The relationship is valid between 15 and 500 m depths. When they applied this relationship to CalCOFI data gathered from 2005 to 2011 (where the data sets are limited to Southern California), the average seasonal values ranged from 8.02 to 8.04 for a depth of 20 m (average of averages is 8.025).

We applied their relationship using our oxygen and temperature data listed in Table 4 and computed the pH values, also listed in that table as pH_est. Our measured values are higher than the values obtained from the empirical relationships. The assumptions behind the empirical relationship rely on the ratios of $CO_2:O_2$ being determined from aerobic activity in water that has not been exposed to the atmosphere for decades. This assumption is probably not valid for coastal Catalina as the large internal waves advect surface-exposed water to depths where pH is measured.

The above results are summarized in Table 6, which includes our bias-corrected measurements. As an aid for comparison, we extrapolated the shallower entries to 18-m depth using a gradient of -0.012 m$^{-1}$, which is the average of the Friedler and CDOC gradients, -0.02 m$^{-1}$ and -0.004 m$^{-1}$, respectively. The CDOC bias-corrected average of depth-profiling and fixed-depth values is 8.095, the largest in Table 6, though matched by the Leinweber and Gruber [19] pH calculations. We conclude that pH at Santa Catalina was the among the highest of sites analyzed and it was similar to the open-ocean values found at SMBO and throughout the Southern California Bight.

## Discussion

Examination of pH modulations or lack thereof may provide insight into the physical processes affecting pH and what pH exposure marine life at the island may experience. First, we consider seasonal variations which are significantly different between pH and temperature.

The temperature variations were qualitatively consistent with our temperature models for the region, with higher surface temperatures and stronger stratification during the summer and fall. When stratification diminishes during the fall, the at-depth temperature increases due to warmer water diffusing from the surface, while the near-surface temperature decreases due to radiational energy loss and the diffusion of colder, deeper water to the surface [15]. The pH depth gradient did align with the temperature stratification, but the near-surface pH did not show seasonal modulation (Fig 13).

Measurement uncertainties limit the amplitude of seasonal variation that we can discern. For example, a seasonal modulation of the near-surface pH is expected based on the annual temperature variation at Santa Catalina as well as seasonal modulations being reported in other studies. The 8 °C [15] seasonal change in surface temperature would produce a 0.056 change in pH due to proton activity as discussed earlier, with the lower pH value occurring during the summer. The uncertainties, number and distribution of our measurements would at best detect unambiguously an annual peak-to-peak variation of 0.06, assuming the mean pH and phase are known.

Somewhat surprising is that biological activity, specifically that of phytoplankton as proxied by chlorophyll, had no discernable seasonal pH signature. Remotely-sensed chlorophyll, as noted by Gelpi [16], is maximal during late fall and early spring. Hence, any pH dependence on $CO_2$ uptake in the bight must be below that assessable with our measurements. Similar statements pertain to published seasonal variations. Hagens and Middelburg [20], have described seasonal variation of pH. They examined seasonal modulations for regions near Iceland, Hawaii and the Mediterranean Sea and compared to other such measurements at various locations. Their studies imply that the typical seasonal modulation they found (~0.02 pH unit) is not discoverable in our data.

This independence of pH (within our measurement uncertainties) with respect to apparent biological activity holds at higher frequencies above the seasonal, i.e., diurnal and semidiurnal frequencies. The correlation of pH modulations with temperature and salinity as well as its semi-diurnal frequency indicate the major pH modulations are abiotic, that is, the modulations are produced by internal waves. In the fixed-depth deployments, high-chlorophyll or high-oxygen content waters advecting past the sensors semi-diurnally were not accompanied by a corresponding high pH signal. Sarmiento and Gruber [21] indicate that every mole of $O_2$ produced during photosynthesis consumes 0.12 mole of $H^+$. As can be seen in both Figs 6 and 10, dissolved oxygen changes by a several tenths of a mg/l near the chlorophyll maximum. If the subsurface dissolved oxygen correlated with the chlorophyll maximum is biologically produced, we expect it to correspond to high pH unless there is significant buffering. Using Fig 10 as an example, there is no pH signal at the depth where the oxygen signal abruptly increases (~18 m).

Also, we did not find a diurnal component to the pH modulation as would be expected if daily $CO_2$ uptake by marine flora was a significant modulator of pH. The power spectral density shown in Fig 3 shows no appreciable power above the background at the diurnal frequency (0.042 hr$^{-1}$). This is in contrast to the findings of Cornwall, *et al.* [22] and Wootton, *et al.* [23], who found a strong diurnal modulation of approximately 0.24 pH units with the maximum at ~15 hours local time. Wootton, *et al.* measured pH in a tidal pool that was connected to the open ocean except during low tide. They attributed the diurnal modulation to daily variation in photosynthesis and respiration. Similarly large diurnal modulations were measured in a *M. pyrifera* kelp forest by Cornwall et al. [22]. Perhaps the local conditions intensified the biological response found in these studies.

We turn to the phenomenology that may produce the results shown in Figs 12 and 13. With no biological forcing, we seek a purely geophysical mechanism. Our speculation is that the

partial pressure of $CO_2$ in the near-surface water was in equilibrium with atmospheric $CO_2$ and its steady value held pH constant there to within our measurement uncertainties. When the surface waters were well-mixed into the water column during the winter, early spring and late fall, this surface pH value was found throughout the mixed layer, yielding small to no depth gradient in pH as noted in Fig 13.

During periods of stratification, i.e., the summer, mixing of surface waters into the water column normally is inhibited and a strong pH gradient is expected. However, these conditions also are conducive to internal wave generation, especially by the semi-diurnal astronomical tides. Due to the nature of internal-wave modulations on a slope as explained by Wunsch [24] and Cacchione and Wunsch [25], and consistent with the temperature modulations shown in Fig 2, surface water temperature is found at depth but deep, cold, low pH water is not brought to the surface. The transport of surface water to depth reduces and perhaps eliminates the pH gradient. The measured gradient would depend on the phase of the internal waves during random deployments and we expect to find a large variation in gradient values during the summer, consistent with the results in the bottom panel of Fig 13.

This mechanism also provides time for the surface waters to equilibrate with the atmosphere. Sarmiento and Gruber [21] argue that the upper-water column equilibrates with the atmosphere on the time scale of days. Hence the near surface would be essentially in equilibrium with the atmosphere. This is consistent with the data shown in Fig 7 in that at higher temperatures (representing near-surface waters) during the deployment the pH was insensitive to temperature.

When internal waves are active, the average pH at depth would be between the nominal at-depth value and the surface value, i.e., average pH is biased by the internal waves. Gelpi [26] computed the pH bias attributed to internal waves to be 0.1 pH unit for the June 2019 fixed-depth deployment. This result is not particular to Santa Catalina as intense internal waves are found both off the mainland and other islands, too. However, it may contribute to higher pH at the island compared to open-ocean sites. Marine life that lives at depths which experience significant internal-wave modulations are subjected to large variations in pH as well as a higher average pH.

However, pH is higher at Santa Catalina relative to the other islands and La Jolla. We believe this is due to less general upwelling in the center of the Southern California Bight where Santa Catalina lies. Upwelling in the center of the bight results from stratification breakdown and subsequent vertical diffusion [15]. This is in contrast to higher California coastal latitudes where northerly winds drive Ekman transport and upwelling, a phenomenon that is usually missing from the center of the Southern California Bight. Also, wind events produce mainland nearshore episodic upwelling [27]. The northern bight (i.e., the Santa Barbara Channel) has its own upwelling mechanisms [28]. The signatures of these phenomena are lower sea surface temperature and increased biological productivity. Average SST shown in Fig 1 confirms that the Northern Channel Islands have lower temperatures compared to the central bight. Remotely-sensed chlorophyll studies [16] confirm that the Northern Channel Islands and the La Jolla region have higher biological activity than Santa Catalina and the central bight.

## Conclusions

The deployment-weighted average pH of the fixed-depth and depth-profile results at Santa Catalina Island in the Southern California Bight are 8.095 at 18.3 m. This value is higher than that obtained from reports of pH at coastal locations within the bight, including the Northern

Channel Islands and the La Jolla kelp forest. However, Santa Catalina pH values are similar to open-ocean values reported for the bight.

There was no apparent seasonal change in pH near the surface. However, we found the pH depth gradient to increase with temperature stratification and the major temporal variation in pH was associated with internal waves, which are seasonally dependent. There was no indication of a biological-activity component to the observed pH modulations. The pH results are consistent with a constant pH at the surface, which during stratification is occasionally advected to depth via internal waves, yielding large variations in the pH depth gradient and increasing the average pH relative to that nominally expected at depth in the absence of internal waves. During well-mixed conditions, higher pH is found throughout the upper water column. The generally higher pH at Santa Catalina appears to be related to less upwelling at the island compared to the northern bight or mainland coast sites.

## Supporting information

**S1 File. Calibration results and salinity gradient.**
(PDF)

**S2 File. Calibration procedure.**
(PDF)

**S1 Dataset.**
(TXT)

**S2 Dataset.**
(TXT)

**S3 Dataset.**
(TXT)

**S4 Dataset.**
(TXT)

**S5 Dataset.**
(TXT)

**S6 Dataset.**
(TXT)

**S7 Dataset.**
(TXT)

**S8 Dataset.**
(TXT)

**S9 Dataset.**
(TXT)

**S10 Dataset.**
(TXT)

**S11 Dataset.**
(TXT)

**S12 Dataset.**
(TXT)

**S13 Dataset.**
(TXT)

**S14 Dataset.**
(TXT)

**S15 Dataset.**
(TXT)

**S16 Dataset.**
(TXT)

**S17 Dataset.**
(TXT)

**S18 Dataset.**
(TXT)

**S19 Dataset.**
(TXT)

**S20 Dataset.**
(TXT)

**S21 Dataset.**
(TXT)

**S22 Dataset.**
(TXT)

**S23 Dataset.**
(TXT)

**S24 Dataset.**
(TXT)

**S25 Dataset.**
(TXT)

**S26 Dataset.**
(TXT)

**S27 Dataset.**
(TXT)

**S28 Dataset.**
(TXT)

**S29 Dataset.**
(TXT)

**S30 Dataset.**
(TXT)

**S31 Dataset.**
(TXT)

**S32 Dataset.**
(TXT)

**S33 Dataset.**
(TXT)

**S34 Dataset.**
(TXT)

**S35 Dataset.**
(TXT)

**S36 Dataset.**
(DAT)

## Acknowledgments

We also thank Karen Norris for technical discussions, planning and logistical support, Desda Sisson and Paul Dimeo of the Aquarium of the Pacific for operations on the Dirk Burcham Scientific Mooring and Jim Updike (American Legion Yacht Club, Newport Beach) for substantial logistical support.

## Author Contributions

**Conceptualization:** Craig G. Gelpi.

**Formal analysis:** Craig G. Gelpi.

**Funding acquisition:** Craig G. Gelpi.

**Investigation:** Craig G. Gelpi.

**Methodology:** Craig G. Gelpi.

**Project administration:** Craig G. Gelpi.

**Software:** Craig G. Gelpi.

**Writing – original draft:** Craig G. Gelpi.

**Writing – review & editing:** Craig G. Gelpi.

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
