## [Decision Letter · Decision Letter 0]

19 May 2023

PONE-D-23-12055Dynamics of pH at Santa Catalina IslandPLOS ONE

Dear Dr. Gelpi,

Thank you for submitting your manuscript to PLOS ONE. After careful consideration, we feel that it has merit but does not fully meet PLOS ONE’s publication criteria as it currently stands. Therefore, we invite you to submit a revised version of the manuscript that addresses the points raised during the review process.

We look forward to receiving your revised manuscript.

Kind regards,

Hui Zhao, Ph.D.

Academic Editor

PLOS ONE

“CDOC was generously supported by the Kenneth T. and Eileen L. Norris Foundation, the Bonnell Cove Foundation and Wet Spot Rentals. The CalCOFI data were obtained from the CalCOFI organization, www.calcofi.org. We also thank Karen Norris for technical discussions, planning and logistical support, Desda Sisson and Paul Dimeo of the Aquarium of the Pacific for operations on the Dirk Burcham Scientific Mooring and Jim Updike (American Legion Yacht Club, Newport Beach) for substantial logistical support.”

“The author received no specific funding for this work.”

4. We note that [Figure 1] in your submission contain [map/satellite] images which may be copyrighted. All PLOS content is published under the Creative Commons Attribution License (CC BY 4.0), which means that the manuscript, images, and Supporting Information files will be freely available online, and any third party is permitted to access, download, copy, distribute, and use these materials in any way, even commercially, with proper attribution. For these reasons, we cannot publish previously copyrighted maps or satellite images created using proprietary data, such as Google software (Google Maps, Street View, and Earth). For more information, see our copyright guidelines: http://journals.plos.org/plosone/s/licenses-and-copyright.

“I request permission for the open-access journal PLOS ONE to publish XXX under the Creative Commons Attribution License (CCAL) CC BY 4.0 (http://creativecommons.org/licenses/by/4.0/). Please be aware that this license allows unrestricted use and distribution, even commercially, by third parties. Please reply and provide explicit written permission to publish XXX under a CC BY license and complete the attached form.

Natural Earth (public domain): http://www.naturalearthdata.com/.

Reviewers' comments:

Reviewer's Responses to Questions

**Comments to the Author**

1. Is the manuscript technically sound, and do the data support the conclusions?

Reviewer #1: Yes

Reviewer #2: Yes

2. Has the statistical analysis been performed appropriately and rigorously? 

Reviewer #1: No

Reviewer #2: No

3. Have the authors made all data underlying the findings in their manuscript fully available?

Reviewer #1: No

Reviewer #2: Yes

4. Is the manuscript presented in an intelligible fashion and written in standard English?

Reviewer #1: No

Reviewer #2: Yes

5. Review Comments to the Author

Reviewer #1: In this study，the author investigated “Dynamics of pH at Santa Catalina Island”. Firstly, the objective is built up to provide a baseline of pH behavior at Santa Catalina Island, USA. Secondly, the author describe the upper water column (to 30-m depth) of pH, temperature, conductivity, chlorophyll and dissolved oxygen at Santa Catalina were made from a fixed mooring and by profiling the water column from a boat and on SCUBA. Thirdly, the author analyze these parameters relationship, and internal waves play an important role on the pH. The last, the pH may be relation to internal waves and upwelling. But the author expound the wide range of pH, and not the verified the materials. Some revisions are as follows：

1、 pH may be relative to the temperature, conductivity, chlorophyll and dissolved oxygen. Would you give the relation between pH and these parameters? That is to say, multivariate statistical analysis should be added.

2、 Internal waves and upwelling should be elaborated in the Introduction.

3、 Results and discussion should be divided into the independent Results and discussion.

4、 What do the Internal waves and upwelling play an important role on pH?

Reviewer #2: Comments on "Dynamics of pH at Santa Catalina Island"

This manuscript studies the pH of the fixed-depth and depth-profile at Santa Catalina Island and analyzes its dynamics. The author found that the pH value is higher than that obtained from reports of pH at coastal locations but similar to open-ocean values reported for the bight. And the author concluded that the pH depth gradient increased with temperature stratification and that the major temporal variation in pH was associated with internal waves. 

As a whole, the manuscript is well arranged. The results showed pH at Santa Catalina Island, which is situated within the Southern California Bight. And the conclusions may be helpful for understanding how pH can vary at specific sites due to oceanographic processes. But the results and discussion are not adequate. There are still some defects and questions in this manuscript. Therefore, I suggest accepting this manuscript if the following questions or errors are well fixed by the author:

(1) Most of the content of the manuscript only displays the observation results, which is more like a note or report than a research article. The manuscript requires some statistical analysis to explain the results. For example, if you want to demonstrate the correlation between temperature, salinity, and pH, you should provide the correlation coefficient between them (Lines 175-177, Lines 188-190, Lines 290-291, etc.). There is too much information in figures 9 and 10, and it is hard to visualize how they relate to each other. It is better to use another figure or table to show the relationship between temperature and pH profiles ("larger pH depth gradients correspond with greater temperature stratification," the sentence in Lines 287-288).

(2) Some abbreviations did not show their full name when they first appeared, i.e., "SCUBA" in the Abstract, "YSI" in Line 78, "SST" in Line 101, "CDOC" in Line 102,... 

(3) There are quite a few sentences that need support from a figure or reference (Lines 174-177, Line 219, etc.); therefore, discussion needs to be further enhanced. In addition, corresponding evidence shall be provided for upwelling and downwelling. 

(4) Lines 66-71, literature review needs to be more specific and targeted. 

(5) Lines 165-167, To say "strong summer stratification conditions" and "well-mixed winter conditions", it is better to give readers the mixed layer depth in summer and winter.

(6) Figure 1, Why choose SST? Land should be marked with other colors. The location of the mooring chain and sampling points should be labeled on the map. 

(7) Lines 174-177, where did you get that? 

(8) It seems that you did not show the relationship between chlorophyll, dissolved oxygen and pH. How did you determine the impact of biological activity on pH?

(9) Lines 230-232, Is the content mentioned above (Lines 160-226) just the extremes? If so, then the conclusion does not have dependability. 

(10) Figure 7, what is the point of this figure? 

(11) Figures 9 and 10 can be combined together.

6. PLOS authors have the option to publish the peer review history of their article (what does this mean?). If published, this will include your full peer review and any attached files.

Reviewer #1: No

Reviewer #2: No

---

## [Author Response · Author response to Decision Letter 0]

14 Jun 2023

I have responded to reviewer and editor comments in the cover letter and the document "Response to Reviewers"

---

## [Decision Letter · Decision Letter 1]

1 Aug 2023

Dynamics of pH at Santa Catalina Island

PONE-D-23-12055R1

Dear Dr. Gelpi,

We’re pleased to inform you that your manuscript has been judged scientifically suitable for publication and will be formally accepted for publication once it meets all outstanding technical requirements.

Kind regards,

Hui Zhao, Ph.D.

Academic Editor

PLOS ONE

Additional Editor Comments (optional):

Reviewers' comments:

Reviewer's Responses to Questions

**Comments to the Author**

1. If the authors have adequately addressed your comments raised in a previous round of review and you feel that this manuscript is now acceptable for publication, you may indicate that here to bypass the “Comments to the Author” section, enter your conflict of interest statement in the “Confidential to Editor” section, and submit your "Accept" recommendation.

Reviewer #1: All comments have been addressed

Reviewer #2: (No Response)

2. Is the manuscript technically sound, and do the data support the conclusions?

Reviewer #1: Yes

Reviewer #2: (No Response)

3. Has the statistical analysis been performed appropriately and rigorously? 

Reviewer #1: Yes

Reviewer #2: (No Response)

4. Have the authors made all data underlying the findings in their manuscript fully available?

Reviewer #1: Yes

Reviewer #2: (No Response)

5. Is the manuscript presented in an intelligible fashion and written in standard English?

Reviewer #1: Yes

Reviewer #2: (No Response)

6. Review Comments to the Author

Reviewer #1: The authors answered the reviewer questions. firstly, the authors had considered the point to point questions, and answeres it step by step. secodly. the authors revised it on corresponding points in the manuscript. that is all.

Reviewer #2: (No Response)

7. PLOS authors have the option to publish the peer review history of their article (what does this mean?). If published, this will include your full peer review and any attached files.

Reviewer #1: No

Reviewer #2: No

---

## [Editor Report · Acceptance letter]

4 Aug 2023

PONE-D-23-12055R1 

Dynamics of pH at Santa Catalina Island 

Dear Dr. Gelpi:

I'm pleased to inform you that your manuscript has been deemed suitable for publication in PLOS ONE. Congratulations! Your manuscript is now with our production department. 

Kind regards, 

on behalf of

Dr. Hui Zhao 

Academic Editor

PLOS ONE